# Quantitative Analysis of the Seminal Plasma Proteome in Secondary Hypogonadism

**DOI:** 10.3390/jcm8122128

**Published:** 2019-12-03

**Authors:** Giuseppe Grande, Federica Vincenzoni, Francesca Mancini, Ferran Barrachina, Antonella Giampietro, Massimo Castagnola, Andrea Urbani, Rafael Oliva, Domenico Milardi, Alfredo Pontecorvi

**Affiliations:** 1International Scientific Institute “Paul VI”, 100168 Rome, Italy; grandegius@gmail.com (G.G.); chicca.mancini@tiscali.it (F.M.); Alfredo.Pontecorvi@unicatt.it (A.P.); 2Divisione di Endocrinologia, Fondazione Policlinico Universitario “Agostino Gemelli” IRCCS, 00168 Rome, Italy; antonella.giampietro@tiscali.it; 3Istituto di Biochimica e Biochimica Clinica, Università Cattolica del Sacro Cuore, 100168 Rome, Italy; fedevincen@gmail.com (F.V.); andrea.urbani@policlinicogemelli.it (A.U.); 4Dipartimento di Scienze di laboratorio e infettivologiche, Fondazione Policlinico Universitario “Agostino Gemelli” IRCCS, 00168 Rome, Italy; 5Molecular Biology of Reproduction and Development Research Group, Institut d’Investigacions Biomèdiques August Pi i Sunyer (IDIBAPS), Department of Biomedical Sciences, Faculty of Medicine and Health Sciences, University of Barcelona, 08036 Barcelona, Spainroliva@ub.edu (R.O.); 6Biochemistry and Molecular Genetics Service, Hospital Clínic, 08036 Barcelona, Spain; 7Laboratorio di Proteomica e Metabolomica, IRCCS Fondazione Santa Lucia, 100168 Rome, Italy; Massimo.Castagnola@unicatt.it

**Keywords:** seminal plasma, proteomics, testosterone, testosterone deficiency, hypogonadism, accessory glands

## Abstract

In the grey zone of testosterone levels between 8 and 12 nmol/L, the usefulness of therapy is controversial; as such, markers of tissue action of androgens may be helpful in adjusting clinical decisions. To better understand the effect of the hypothalamic-pituitary-testicular axis on male accessory secretion, we performed a proteomic quantitative analysis of seminal plasma in patients with secondary hypogonadism, before and after testosterone replacement therapy (TRT). Ten male patients with postsurgical hypogonadotrophic hypogonadism were enrolled in this study, and five of these patients were evaluated after testosterone treatment. Ten men with proven fertility were selected as a control group. An aliquot of seminal plasma from each individual was subjected to an in-solution digestion protocol and analyzed using an Ultimate 3000 RSLC-nano HPLC apparatus coupled to a LTQ Orbitrap Elite mass spectrometer. The label-free quantitative analysis was performed via Precursor Ions Area Detector Node. Eleven proteins were identified as decreased in hypogonadic patients versus controls, which are primarily included in hydrolase activity and protein binding activity. The comparison of the proteome before and after TRT comes about within the discovery of six increased proteins. This is the primary application of quantitative proteomics pointed to uncover a cluster of proteins reflecting an impairment not only of spermatogenesis but of the epididymal and prostate epithelial cell secretory function in male hypogonadism. The identified proteins might represent putative clinical markers valuable within the follow-up of patients with distinctive grades of male hypogonadism.

## 1. Introduction

Male hypogonadism is characterized as “inadequate gonadal function, as manifested by deficiency in gametogenesis and/or secretion of gonadal hormones” [1]. It can be classified as primary, secondary, or tertiary hypogonadism according to the site primarily affected (the testis, the pituitary gland, or the hypothalamus, respectively). Hypogonadotrophic hypogonadism is characterized by low testosterone (T) plasma levels combined with normal or low FSH and LH plasma levels, due to an impairment of the pituitary function. The diagnostic protocol of secondary hypogonadism, also known as hypogonadotropic hypogonadism (HH), includes an accurate medical history, a physical exam, semen analysis, hormone measurements, and instrumental evaluation.

Total testosterone levels of less than 8 nmol/L profoundly bolster a determination of hypogonadism whereas levels greater than 12 nmol/L are likely to be typical. An inadequate androgen status is possible if the total testosterone levels are between 8 and 12 nmol/L. The diagnosis of hypogonadism and the appropriateness of a treatment in the “grey zone” between 8 and 12 nmol/L is debated and should be supported by symptoms, often non-specific. Most patients with a total testosterone between 8 and 12 nmol/L, if asymptomatic, will not be hypogonadal. Patients experiencing symptoms of hypogonadism require further investigations [2]. To date, no markers of androgen tissue action have been suggested and validated in clinical practice. As such, new markers of androgens action might be valuable in orienting diagnosis, decision about testosterone replacement treatment (TRT), and clinical follow-up.

As in all endocrine diseases, the treatment goal for HH is to reestablish the lacking glandular work. Several studies reported that gonadotropin (hCG and FSH) therapy is effective in initiating and maintaining spermatogenesis and moreover should be preferred to testosterone for inducing an increase in vitamin D levels and for being associated with lower concentrations of estrogens [3]. Because of their greater expense and complexity, however, gonadotropins are usually reserved for men with gonadotropin deficiency who desire fertility and in whom spermatogenesis must be initiated and maintained [4]. If fertility is the topic, FSH and LH administration is in fact recommended. In all the other conditions, testosterone replacement therapy (TRT) is the most helpful choice [5,6]. The goals of TRT are to restore the T levels in serum within the mid-normal physiological range associated with the patient’s age group, generally considered to be between 13.8 and 24.27 nmol/L, and to improve symptoms in hypogonadal men [7]. Although most of the positive effects of TRT begin to occur between 3 to 6 weeks after initiation, considerable variations have been observed and some patients require up to 1 year to observe clinical effects [8].

Previous data have been provided about the effect of T deficiency and TRT on male fertility, in terms of function of male accessory sex glands (epididymis, prostate, and seminal vesicles). Specifically, it has been reported the effect of orchiectomy on epididymal function. In rats, orchiectomy actuates a diminishment in epididymal weight that is less checked than for prostate or seminal vesicles [9]. Prostate is a highly androgen-dependent tissue. The androgens play a fundamental part in prostate growth and development and within the pathogenesis of prostate diseases, such as benign prostatic hyperplasia (BPH) and adenocarcinoma [10]. In 2009 Ma et al., using a transcriptomic approach, reported the identification of 187 transcripts significantly affected by dihydrotestosterone (DHT) reduction in mice prostate [11]. This finding underpins the speculation that androgen-regulated genes are involved in prostate function [11]. However, few studies have been focused on the androgen action mechanism in epididymal epithelial cells or in prostate tissue, likely due to the need for adequate tools.

One of the opportunities to further elucidate molecular mechanisms of androgen modulation of accessory gland function is offered by proteomics, which captures the overall protein profile instead of the expression of individual genes. Due to the dynamic nature of the proteome, proteomic-based personalized medicine is fluid, adapting to individuals and individual situations [12]. Recent advances in proteomic techniques, proteomic data analysis, and application of proteomic techniques in clinical settings undoubtedly represent a real promise for early disease diagnosis, prognosis, and theragnosis on an individual basis. There is a small question that the next decade will be the time of proteomics [13]. In the area of male fertility, proteomics appears the foremost promising and effective platform recently applied in order to widely study the physiology and pathophysiology of male reproduction [14].

In 2014, we published the first pilot proteomic study, applying high-resolution mass spectrometry before and after 6 months of TRT in human seminal samples of patients affected by hypogonadotropic hypogonadism. We reported the identification of a panel of 14 absent proteins in hypogonadic patients with respect to normogonadic subjects [15].

However, no quantitative proteomic studies have yet been performed to understand how testosterone modulates protein abundance and how the protein abundance is quantitatively regulated by testosterone.

Recent years have in fact witnessed a significant improvement of mass spectrometers. Orbitrap MS was recently improved combining a dual-pressure linear ion trap (Velos Pro) with a new high-field Orbitrap™ mass analyzer to create the ultimate analytical instrument (Orbitrap Elite) [16], which was used in the present study. Furthermore, bioinformatics is more and more capable of supporting proteomics in the interpretation of the data analysis and in the identification of the proteins of clinical interest by means of statistical analysis and use of algorithms [17]. As a consequence high-resolution, quantitative, proteomic approaches may now be useful in resolving clinical questions, such as in this case, the identification of quantitatively regulated proteins by testosterone in seminal plasma or the identification of early markers of response to TRT.

With this aim, we performed a quantitative high-resolution proteomic analysis in seminal plasma samples of patients affected by HH, before and after only 3 months of TRT.

## 2. Materials and Methods

The study design was approved by the Ethical Committee of Fondazione Policlinico “A. Gemelli”, Rome (Italy). A written informed consent in accordance with the Declaration of Helsinki has been given by all subjects.

### 2.1. Human Subjects

Ten male patients aged between 25 and 55 years with postsurgical HH were selected for this study.

No patients displayed hypogonadism when evaluated before the neurosurgical operation at endocrinological clinic of our tertiary care university hospital (“A. Gemelli”, Rome, Italy). All patients underwent andrological evaluation at our clinic 6 months after the neurosurgical operation, which removed a pituitary adenoma or craniopharyngioma. During this time, replacement therapies for thyroidal, adrenal, and somatotropic axes were performed when single or multiple deficits were documented. All patients were also assessed during this time to exclude residual adenoma.

The rationale for studying only men with HH was to select a condition of extreme reduction in blood T levels without other confounding variables. The patients with primary hypogonadism were excluded because their blood T is usually not as low as that of subjects with HH. Moreover, patients with primary hypogonadism often show normal or increased serum estrogen levels, which might represent a confounding factor. Inclusion criteria were as follows: total T less than 8 nmol/L, calculated free T less than 1.6%, and clinical symptoms of hypogonadism. Exclusion criteria for the study included age younger than 20 years and older than 55 years, primary hypogonadism or associated testicular diseases, smoking, residual adenoma, previous androgen replacement therapy, diabetes mellitus, varicocele, and genital tract infections.

Five patients were also evaluated after 3 months of T replacement therapy (TRT) with transdermal testosterone (Tostrex 2% gel), to assess the early impact of androgen substitution treatment on seminal proteome. The evaluation has been performed after 3 months of TRT because it is the proper timing of response to exogenous androgen administration in terms of restoration of seminal vesicle size, which is a sensitive measure of androgen genomic effect [18].

Ten fertile men, whose partners were pregnant when the study started, were selected as a control group in the study. None of them had a history of infertility. All female partners conceived within 3 months before the start of the study.

### 2.2. Hormonal Study

A blood sample was collected at admission at the andrological clinic 6 months after the neurosurgical operation, at 8:00 a.m. for the determination of sex hormone-binding globulin (SHBG), T, estradiol (E2), luteinizing hormone (LH), and follicle-stimulating hormone (FSH). T and E2 were measured in duplicate by radioimmunoassay (RIA) with the use of commercial kits (Radim, Pomezia, Italy). LH, FSH and SHBG were assayed by immunoradiometric methods on a solid-phase (coated tube), which is based on a monoclonal double-antibody procedure. Reference values of the studied hormones are reported in Table 1.

Testosterone and estradiol levels were then measured in patients receiving TRT after 3 of treatment to confirm the effectiveness of TRT.

The intra-assay coefficients of variation (percentage) were 6.1% for T, 2.3% for E 26.9% for SHBG, 5.6% for LH, and 6.9% for FSH. The inter-assay coefficients of variation were 9.3% for T, 3.5% for E2, 8.5% for SHBG, 9.1% for LH, and 8.4% for FSH.

### 2.3. Semen Analysis and In-solution Digestion

Complete semen analysis was performed upon admission at the andrological clinic in controls, and in patients 6 months after the neurosurgical operation, according to WHO (2010) classification [19].

Liquefied semen samples were then centrifuged at 3000× *g* for 30 min to get the seminal plasma and to ensure complete expulsion of the cellular components. After the centrifugation, an aliquot was checked under a microscope to confirm that no spermatozoa were displayed. Seminal plasma was divided in 0.5 mL aliquots, which were quickly frozen at −80 °C until mass spectrometry (MS) examination was carried out inside 1 month.

Seminal plasma samples were subjected to solution digestion. Briefly, an aliquot of seminal plasma corresponding to 50 µg of total protein (as measured by Bradford assay) was mixed with 100 mM ammonium bicarbonate pH 8.0 and reduced with 200 mM dithiothreitol (DTT; 10 mM final concentration, Sigma-Aldrich, St. Louis, MI, USA) for 5 min at 100 °C and 15 min at 50 °C, and alkylated with 200 mM iodoacetamide (IAA; 55 mM final concentration, Sigma) in the dark at room temperature for 60 min. The samples were left to digest overnight at 37 °C by adding ammonium bicarbonate solution with sequencing grade modified porcine trypsin (1:50, trypsin: protein concentration, Promega, Madison, WI, USA). To stop the digestion, the samples were acidified with aqueous trifluoroacetic acid (TFA/H_2_O 0.2% (*v*/*v*)) and immediately frozen and lyophilized.

### 2.4. Proteomic Analysis

For proteomic analysis, the samples were suspended in aqueous formic acid [FA/H_2_O 0.1% (*v*/*v*)] and equal protein quantity (5 µg) of each sample was analyzed using an Ultimate 3000 RSLCnano HPLC System coupled to a LTQ Orbitrap Elite mass spectrometer (ThermoFisher). Separation was performed using a Zorbax 300SB-C18 column (3.5 mm particle diameter; column dimension 1 mm i.d. 15 cm, Agilent Technologies) and the following eluents: (A) 0.1% (*v*/*v*) aqueous FA and (B) acetonitrile: H_2_O 80:20 with 0.1% (*v*/*v*) aqueous FA. We applied a linear gradient from 0 to 55% of solvent B in 60 min, at a flow rate of 50 μL/min. The Elite-Orbitrap mass spectrometer was operated in data-dependent mode in which each full MS scan (60,000 resolving power) was followed by MS/MS scans where the five most intense multiple-charged ions were dynamically selected and fragmented by collision-induced dissociation (CID) at a normalized collision energy of 35% and acquired in linear ion trap at normal scan rate. Samples were analyzed individually; proteomic analysis was performed at the same time for all samples, while data analysis was subsequently performed.

### 2.5. Data Analysis

Tandem mass spectra were analyzed using the Thermo Proteome Discoverer 1.4.1.14 software based on SEQUEST HT cluster as the search engine (University of Washington, Seattle, WA, USA, licensed by Thermo Electron Corp, Waltham, MA, USA) against *Homo sapiens* UniProtKb/Swiss-Prot protein knowledgebase (release date: 2017-02).

Data were searched for two missed cleavages, cysteine carbamidomethylation as a static modification, and methionine oxidation as a dynamic modification. Criteria utilized to accept protein identification included a false discovery rate (FDR) of 1% and at least 1 unique peptide match per protein. The dissociated or ‘ungrouping’ of proteins from their respective families was used during the quantification process to avoid the possible ambiguity associated with different isoforms of the same protein.

The label-free quantitative analysis was performed via Precursor Ions Area Detector Node during the bioinformatic analysis using Proteome Discoverer software. This quantification method was used to define the relative quantities of all peptides in a sample. The Proteome Discoverer application calculates peptide areas during processing, using them to automatically calculate protein areas for the proteins in the report. It calculates the area of any given protein as the average of the three most abundant distinct peptides identified in the protein.

Mean ± standard deviation of protein abundance was calculated for each protein in the group of controls (*n* = 10) and in the group of hypogonadic patients (*n* = 10). The relative protein level ratios between the group of controls and the group of hypogonadic patients (*n* = 10) were determined from the respective averages of protein abundances expressed in all patients. We compared for each protein the mean abundance in the group of HH patients with the one reported in the group of healthy males. All the proteins detected with a ratio > 1.5 (less abundant proteins in HH patients) or <0.67 (more abundant proteins in HH patients) have been considered for this study.

In the population of 5 patients who received TRT, we moreover compared, for each patient, the protein abundance in samples obtained before and after TRT therapy and determined the pre-treatment: post-treatment ratio, obtaining a list of under-expressed (ratio > 1.5) or over-expressed (ratio < 0.67) proteins after TRT.

The cut-off values of 1.50 and 0.67 for mean abundance ratios have been selected as previously reported in literature [20,21,22].

Proteins identified by SEQUEST were then analyzed using the publicly available protein annotation through evolutionary relationship (PANTHER) classification system (http://www.pantherdb.org/). Furthermore, we evaluated the Gene Ontology Molecular Function annotations in the list of the proteins differently expressed in patients versus controls.

### 2.6. Western Blot

In order to validate the proteomics results, a western blot analysis was performed in seminal plasma samples used for proteomics. Seminal plasma samples used for confirmation analysis included the 10 hypogonadic patients, and 8 of the 10 controls, since 2 control samples were spent for proteomic analysis. Furthermore, to increase the power of the confirmation analysis we added in western blot analysis 7 independent hypogonadic patients samples, furtherly included in the study with the same inclusion and exclusion criteria. Clinical and hormonal parameters of this population of 7 additional hypogonadic patients is reported in Appendix A. To summarize, the western blot confirmation analysis was performed on seminal plasma samples of 17 hypogonadic patients and 8 controls.

For western blot analysis, seminal plasma were diluted in RIPA buffer (50 mM Tris–Cl, pH 7.5, 150 mM NaCl, 1% Nonidet P-40, 0.5% Na desoxicholate, 0.1% SDS, 1 mM EDTA) supplemented with a cocktail of protease inhibitors (Boehringer, Ingelheim am Rhein, Germany). Samples were sonicated, quantified by Bradford protein assay, resuspended in SDS Laemmli sample buffer, and boiled for 5 min at 95 °C. Proteins (30 µg) were separated by SDS–PAGE and subsequently transferred onto PVDF membranes (Millipore, Burlington, MA, USA). Membranes were developed using the enhanced chemiluminescence (ECL Amersham, Little Chalfont, United Kingdom). The following primary antibodies were used: anti-Lactoferrin polyclonal antibody (1:2000) and anti-PIP polyclonal antibody (1:500) (Novus Biologicals, Centennial, CO, USA). Red ponceau staining was used as internal control to check the equal loading of total proteins in western blot analysis [23].

Values were reported as the mean ± S.D. Statistical analysis was performed by the two-tailed unpaired Student’s *t*-test using the SigmaStat 4.0 software (Systat Software Inc., San Jose, CA, USA). Differences were considered statistically significant when *p* < 0.05.

## 3. Results

The results derived from the clinical, hormonal, and seminal data of the 10 hypogonadic patients and the 10 controls are reported in Table 1.

The mean age of the patients in the two groups did not differ significantly. Total T, FSH, and LH resulted significantly decreased in hypogonadic patients compared with the control group. Seminal volume, sperm count, sperm motility, and normal sperm morphology were significantly reduced compared to hypogonadic patients vs. controls. The T levels in hypogonadic patients were restored after TRT treatment as reported in Table 2.

We identified—before stringent filters were applied—110 to 175 proteins per individual subject sample in the hypogonadic group, and 117 to 176 proteins in the control group. Furthermore, since samples had not been depleted, semenogelins resulted in the majority of the identified spectra.

As a consequence, the adoption of stringent protein identification criteria (FDR of 1% and at least 1 unique peptide match per protein) resulted in the characterization of 12–17 proteins per individual subject sample in the hypogonadic group and 15–60 in the control group which have been identified with high accuracy and, therefore, finally considered for our study. The comparison of the seminal plasma proteomes of the hypogonadic patients and control groups resulted in the detection of 11 differentially expressed proteins (ratio controls/HH patients < 0.67 or > 1.5; Table 3). Of those, no proteins were found only in the group on HH patients or at higher levels in the group of patients (ratio < 0.67), whereas all the identified 11 proteins were found at lower levels (ratio > 1.5).

The analysis of GO annotations for Molecular function of the 11 differentially expressed seminal proteins revealed that the most frequent molecular functions were catalytic activity (among the catalytic proteins the major part were involved in hydrolase activity) and binding (among the binding proteins the major part were involved in protein binding activity).

The comparison of the proteomes of the hypogonadic patients before and after TRT resulted in the detection of 6 differentially expressed proteins, which were found to be increased after TRT (ratio after/before TRT > 1.5; Table 4).

No proteins have been observed as reduced after TRT.

Western blot analysis confirmed proteomic data and namely the significant reduction in PIP and lactoferrin levels in hypogonadic patients versus controls (Figure 1).

In the Appendix A, we report the picture of all gels after enhanced chemiluminescence, all pictures of membranes after red ponceau staining and western blot membrane with detected bands. Quantification has been furthermore reported as Appendix A.

## 4. Discussion

In this study, we took advantage of the application of high-resolution mass spectrometry technology and a quantitative proteomic approach to decide the in vivo effect of androgen lack on male accessory sexual glands. The full list of the 11 identified decreased proteins in hypogonadism may allow the development of an array of proteins related to T action on normal sexual glands function. From a fundamental perspective it also points to the in vivo tissue response of male accessory sexual glands to testosterone deficiency. Furthermore, the quantitative modulation in the abundance of these proteins, although needing further confirmation, may be used to identify new markers useful in the diagnosis of male hypogonadism, especially in patients with T levels in the “grey zone” between 8 and 12 nmol/L.

The proteins reported as reduced in male hypogonadism are mainly involved in catalytic activity, and namely in hydrolase activity, thus suggesting a homeostasis disorder between proteases and their inhibitor in semen of HH patients, and in protein binding activity. Specifically, the proteins identified with hydrolase activity are plasma serine protease inhibitor (Serpina 5 or IPSP) and gastricsin (PEPC). Serpina 5, also designated as Protein C inhibitor (PCI) or plasminogen activator inhibitor 3, is a member of the serine protease inhibitor (serpin) family that inactivates serine proteases by forming stable, enzymatically inactive enzyme inhibitor complexes [24]. PCI inhibits many proteases such as activated protein C [25,26], thrombin [27], factor Xa [27], factor Xia [27,28], plasma kallikrein [27,28], thrombin–thrombomodulin complex [29], urokinase (uPA) [30,31], tissue plasminogen activator (tPA) [27,31], the sperm protease acrosin [32,33], tissue kallikrein [34,35], and prostate specific antigen (PSA or KLK3) [35,36]. Disruption of the PCI gene, which is highly expressed in the male reproductive tract, comes in infertility of male homozygous PCI-knockout mice [37]. Spermatozoa derived from PCI^-/-^ males were malformed and were not able to bind and to fertilize oocytes, as shown by in vivo and in vitro fertilization experiments [37]. Histological analysis of PCI^-/-^ mice revealed abnormal spermatogenesis associated with damage of Sertoli cells [37]. Apart from the importance of PCI in spermatogenesis and spermiogenesis, previous immunohistochemistry studies together with the malformation of spermatozoa observed in PCI^-/-^ deficient mice show that PCI may play an important role in further sperm cell maturation during the passage through the epididymis as deduced from the presence of PCI in the cytoplasmic droplet (CD) [24]. Serpina 5 is moreover secreted in seminal plasma, where it inhibits many proteases, which plays pivotal roles in male reproduction [35]. The reduction of PCI in hypogonadic patients might reflect an impairment in spermatogenesis or might represent a signature of defective spermatogenesis in HH patients, as well as in epididymal function. Further studies are needed to confirm these hypotheses.

On the other hand, Gastricsin, which in seminal plasma is a prostate-derived protein, was reported to be involved in the degradation of many seminal plasma proteins at low pH in vitro [38]. This role of gastricsin was subsequently confirmed by in vivo studies [39], where it was demonstrated that gastricsin in seminal plasma is activated at low pH in the vagina 2–7 h post-coitus and that gastricsin activity is present more than 24 h thereafter [38]. Thus, gastricsin-mediated cleavage of seminal plasma proteins is a documented in vivo phenomenon. Further studies demonstrated that epididymal protein hCAP-18 is cleaved in vagina by gastricsin into two parts: the cathelin part and ALL-38, which has an antimicrobial activity against a variety of microorganisms [40]. Consequently, in hypogonadic patients, we reported a reduction in this defense mechanism against infection transmission during sexual intercourse. Previous data have been reported for HH patients about the higher incidence in these patients of prostatitis and male tract infection/inflammation (MAGI) [41] and the increased levels of MAGI markers in seminal plasma [42]. The observation of a reduction in gastricsin in our study might represent a molecular mechanism by which hypogonadic patients have more frequently male tract infections. Further studies are needed to confirm this hypothesis.

Binding proteins found to be reduced in hypogonadic patients in our study include beta-microseminoprotein (MSMB), plasma serine protease inhibitor (IPSP) and lactotransferrin (TRFL). MSMB, also known as PSP94, is secreted by the epithelial cells of the prostate [43] and is one of the major constituents display in human seminal plasma [44]. A significant spontaneous acrosome reaction inhibition was observed in a model of guinea pig when epididymal spermatozoa were pre-incubated with MSMB [45]. In addition, previous studies have shown that MSMB binds human immunoglobulin [46,47]. It has been hypothesized that high amounts of MSMB present in the seminal plasma would bind to immunoglobulin and prevent an immune response to spermatozoa in the female reproductive tract [46]. MSMB has been moreover shown to interact with prostatic acid phosphatase (PPAP) and CRISP-3 proteins present in seminal plasma [48]. Similarly, a PPAP-containing zinc-binding multiprotein complex has also been characterized from the human seminal plasma [49], and we previously reported that male hypogonadism is associated with a reduction in seminal PPAP levels [15]. Here we report the reduction of both PPAP and MSMB in seminal plasma of male hypogonadism. The role of PPAP and MSMB as a part of such a functional network needs to be further examined.

Furthermore there is lactoferrin, which has an antioxidative, antibacterial, and immune-modulating role in seminal plasma [50]. Lactoferrin is additionally included in keeping up typical sperm structure and motility, and in tweaking the composition and quality of the semen during sperm maturation and migration through the genital tract [51]. We previously reported that male hypogonadism actuates a noteworthy decrease in lactoferrin [15]. Here, we confirm, through a quantitative proteomic approach, our previous observation.

Interestingly, the effect of testosterone replacement therapy in our hypogonadic patients has induced an increase in 6 proteins (Table 3). Besides, it is important to highlight that SEMG1, SEMG2, PPAP, PIP and TRFL protein levels have been observed as significantly reduced in hypogonadic patients versus controls, but their levels were increased after TRT (Table 2 and Table 3), representing a putative panel for the diagnosis of male hypogonadism and the follow-up of the treatment. SEMG1, SEMG2 and PPAP are known androgen-dependent secretory products [52,53]. Of note, PIP is an aspartyl proteinase that ties to numerous proteins, counting human zinc-α-2 glycoprotein [54]. Its capacity to tie an expansive cluster of proteins demonstrates its multifaceted part in various biological processes, such as fertility, antimicrobial activity and immune-regulation [55]. A decrease of seminal PIP was moreover described in asthenospermic patients [56]. We reported in 2014 that PIP was not detectable by qualitative proteomics in seminal plasma by hypogonadic patients [15]. Here we confirm this observation, as PIP was found at lower levels in hypogonadic patients, and, therefore, we suggest that PIP might represent one of the most useful markers for male hypogonadism diagnosis and TRT follow-up.

This study represents the primary application of quantitative high-resolution MS-based proteomics pointed to distinguish an array of proteins in the seminal plasma of hypogonadic patients and reflecting a disability of epithelial cell secretory function in male secondary hypogonadism.

Some limitations of this study are that it was performed on a relatively small sample size. However, the detected proteins and its function are consistent with previous data described in the literature, as discussed above. It was moreover performed in a population receiving only TRT, not compared with patients treated with hCG. Further studies are so needed comparing patients treated with TRT and patients treated with hCG.

Furthermore, the identification of proteins involved in androgen action may represent the basis to clarify how androgens act in the physiology of the sexual accessory glands and could help to explain the link between hypogonadism and infertility. The proteomic approach performed herein in seminal plasma permitted to identify novel molecular markers, which might be translated in clinical practice for the diagnosis of patients with distinctive grades of male hypogonadism and in the TRT follow-up.

## Figures and Tables

**Figure 1 jcm-08-02128-f001:**
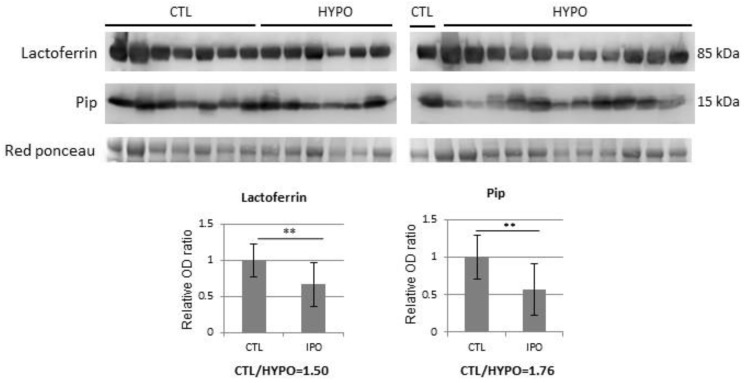
WB analysis of Lactotransferrin (TRFL) and Prolactin-inducible protein (PIP) proteins in seminal plasma samples from controls (CTL) and hypogonadic patients (HYPO). Histograms show the ratio of densitometric values of Lactoferrin and PIP to Red Ponceau. The ratio from controls (CTL) was arbitrarily set to 1. Mean ± SD of patients is shown (** = *p* < 0.01, two-tailed unpaired *t*-test).

**Table 1 jcm-08-02128-t001:** Clinical and hormonal parameters in controls and hypogonadotropic hypogonadic (HH) patients—proteomic analysis of seminal plasma; * *p* < 0.05.

	Hypogonadic Patients (*n* = 10)	Controls (*n* = 10)	Range Values
Age	37.5 ± 9.1	39.6 ± 8.3	
Testosterone (T)	1.87 ± 0.64	4.9 ± 0.9 *	2.5–8.4 ng/mL
Estradiol (E2)	23.25 ± 4.72	27.9 ± 9.4	15–44 pg/mL
SHBG	19.67 ± 4.34	38.2± 9.3	16–80 nmol/L
FSH	1.72 ± 1.14	2.6 ± 1.3 *	1.0–8.0 mU/mL
LH	0.90 ± 0.50	2.9 ± 0.8 *	2.5–10.0 mU/mL
Seminal volume	1.75 ± 1.13 mL	2.9 ± 0.9 *	
Sperm concentration	2.61 ± 4.83 × 10^6^/mL	74.2 ± 28.3 × 10^6^/mL *	
Total sperm motility	8.3 ± 12.90%	52.6 ± 8.3% *	
Normal morphology	3.2 ± 4.1%	8.9 ± 2.0% *	

**Table 2 jcm-08-02128-t002:** Clinical and hormonal parameters in hypogonadotropic hypogonadic (HH) patients before and after testosterone replacement therapy; * *p* < 0.05.

	After TRT (*n* = 5)	After TRT (*n* = 5)	
Testosterone (T)	1.24 ± 0.76	3.63 ± 0.48 *	2.5–8.4 ng/mL
Estradiol (E2)	21.47 ± 5.72	29.48 ± 8.73	15–44 pg/mL

**Table 3 jcm-08-02128-t003:** Reduced seminal proteins in hypogonadotropic hypogonadic patients (HH; *n* = 10) versus controls (C; *n* = 10).

Accession Number	Protein Description	Gene	Ratio C/HH
P05154	Plasma serine protease inhibitor	IPSP	2.89
P12273	Prolactin-inducible protein	PIP	2.32
P54107	Cysteine-rich secretory protein 1	CRIS1	2.21
P20142	Gastricsin	PEPC	2.19
P04279	Semenogelin-1	SEMG1	2.12
P08118	Beta-microseminoprotein	MSMB	2.04
Q02383	Semenogelin-2	SEMG2	1.92
P07288	Prostate-specific antigen	KLK3	1.65
P07602	Prosaposin	SAP	1.64
P15309	Prostatic acid phosphatase	PPAP	1.54
P02788	Lactotransferrin	TRFL	1.52

**Table 4 jcm-08-02128-t004:** Increased seminal proteins in hypogonadic patients after TRT (Ratio before/after TRT > 1.5; *n* = 5).

Accession Number	Protein Description	Gene	Ratio After/Before TRT
P04279	Semenogelin-1	SEMG1	2.42
Q02383	Semenogelin-2	SEMG2	2.15
Q6W4X9	Mucin-6	MUC6	1.87
P12273	Prolactin-inducible protein	PIP	1.76
P15309	Prostatic acid phosphatase	PPAP	1.64
P02788	Lactotransferrin	TRFL	1.51

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
