# Peer review of "Quantitative Analysis of the Seminal Plasma Proteome in Secondary Hypogonadism"

_jcm, 2019, doi:10.3390/jcm8122128_

Round 1

Reviewer 1 Report

The main limit of this study is represented by clinical design. Because patients with HH do not receive treatment with hCG ?

Another problem is represented by the small number of analyzed patients.

In the clinical practice is not plausible suggest treatment with testosterone for patients in fertile age, therefore the authors should provide data after treatment with hCG. If not possible, this aspect must be reported as limitation of the study.

At this moment this article deserves major revision.

Author Response

We are grateful to the Reviewers for their observations.

Reviewer1:

The main limit of this study is represented by clinical design. Because patients with HH do not receive treatment with hCG ?

Another problem is represented by the small number of analyzed patients.

In the clinical practice is not plausible suggest treatment with testosterone for patients in fertile age, therefore the authors should provide data after treatment with hCG. If not possible, this aspect must be reported as limitation of the study.

At this moment this article deserves major revision.

Reply: Thanks for your suggestions. We reported the absence of a group treated with hCG as one of the limits of the study. Furthermore we briefly discussed this topic in the introduction section. In fact, several studies reported that gonadotropin (hCG and FSH) therapy is effective in initiating of maintaining spermatogenesis and moreover should be preferred to testosterone for inducing an increase in vitamin D levels and for being associated with lower concentrations of estrogens. However, because of their greater expense and complexity, gonadotropins are usually reserved for men with gonadotropin deficiency who desire fertility and in whom spermatogenesis must be initiated and maintained. If fertility is the topic, FSH and LH administration is in fact recommended. In all the other conditions, testosterone replacement therapy (TRT) is the most most helpful choice.

We moreover understand that 10 patients seem a small number for a clinical study; it is a sufficient number however for proteomic analysis, since it is an expensive and time-consuming technique. Further clinical studies are needed to confirm the data suggested by this proteomic report on larger populations of HH patients.

Reviewer 2 Report

The authors of the paper “Quantitative analysis of the seminal plasma proteome in secondary hypogonadism” performed a quantitative proteomic analysis of i) seminal plasma originated from patients affected by hypogonadism (HH) in comparison to healthy men (controls), and ii) seminal plasma proteome of patients before and after testosterone replacement treatment (TRT).

The introduction of mass spectrometry to analysis of seminal plasma proteome of HH patients and TRT was described by Milardi et al., 2014 (J Clin Endocrinol Metab. doi: 10.1210/jc.2013-4148) with very similar experimental design. Therefore, I wonder about novelty aspect of present study. The innovative nature of research should be highlighted

The authors must clearly identify a problem or topic that they want to explore. First, what is the reason for quantitative proteomic analysis between HH and healthy men, and second for comparison patients before and after TRT. Materials and methods should be written in more details. The results seems to be incomplete and at least some hypotheses should be tested.

In my opinion, the English should be corrected.

Introduction

L80: please be more precise and explain the molecular mechanism of what you thought about?

L95: the aim of study should be more precise

Material and methods

Experimental design is not clear. It should be clearly indicated which experimental groups were compared e.g. seminal plasma proteome of HH males with healthy males, and seminal plasma proteome of HH males before and after 3 month of TRT.  Why 3 month period of treatment was chosen. Preciously the authors studied the effect of 6 months of TRT (Milardi et al., 2014; doi: 10.1210/jc.2013-4148. Epub 2014 May 5)

L120: Was the effectiveness of TRT measured?

L182-191 Was the abundance of identified proteins significantly different?

L186, 190, 191: why different ratio was applied for comparison HH and healthy males (ratio>1.5 less abundant for HH and <0.67 more abundant for HH) and before and after TRT (ratio>1.5 under-expressed after TRT and <0.67 over-expressed after TRT). Usually ratio below above 2 is considered for proteomics.

L201: there is no information about additional 7 clinical and hormonal hypogonadic patients samples used for western blot.

L205-213: amount of protein loaded on SDS-PAGE is unknown.

L212-213: please add reference for red ponceau staining as internal standard

L196-213: there is lack of description of statistical method that were used for analysis of data presented in Fig.1

Results

Table 1. there are only results concerning hypogonadic patients and control. Why authors have not provided the same data for patients before and after TRT.

L227: what does it mean  "the adoption of stringent protein identification criteria". Please, describe all criteria in details in M&M section.

It means that from more than 100 identified proteins (L224 and 225) only 11 was differentially expressed between HH and healthy patient. And all of them was lower for HH, there was any protein characteristic for HH?

The same for table 3. Any proteins with higher abundance before treatment.

Fig. 1. In the supplementary material I would like to see the picture of all gels after enhanced chemiluminescence, all pictures of membranes after red ponceau staining and western blot membrane with detected bands. Please show the calculation and  complete the description of the charts (Y axis is empty).

Discussion

L:263-283. Do you suggest  homeostasis disorder between proteases and their inhibitor in HH patients?

L274: You mentioned about malformed sperm for mice, could you check your hypothesis (L282-283) for HH patients?

L292-293, could you support your hypothesis concerning reduction of defence mechanism against infection transmission during sexual intercourse in HH patients.

L318. To be sure that the level of some proteins were restored you should compare HH patients after TRT with control. Now you can only suggest the increase of proteins after TRT.

Author Response

We are grateful to the Reviewers for their observations.

Reviewer2:

The authors of the paper “Quantitative analysis of the seminal plasma proteome in secondary hypogonadism” performed a quantitative proteomic analysis of i) seminal plasma originated from patients affected by hypogonadism (HH) in comparison to healthy men (controls), and ii) seminal plasma proteome of patients before and after testosterone replacement treatment (TRT).

The introduction of mass spectrometry to analysis of seminal plasma proteome of HH patients and TRT was described by Milardi et al., 2014 (J Clin Endocrinol Metab. doi: 10.1210/jc.2013-4148) with very similar experimental design. Therefore, I wonder about novelty aspect of present study. The innovative nature of research should be highlighted

The authors must clearly identify a problem or topic that they want to explore. First, what is the reason for quantitative proteomic analysis between HH and healthy men, and second for comparison patients before and after TRT.

Experimental design is not clear. It should be clearly indicated which experimental groups were compared e.g. seminal plasma proteome of HH males with healthy males, and seminal plasma proteome of HH males before and after 3 month of TRT. Why 3 month period of treatment was chosen. Preciously the authors studied the effect of 6 months of TRT (Milardi et al., 2014; doi: 10.1210/jc.2013-4148. Epub 2014 May 5)

Reply: We thank the Reviewer for permitting us to specify the innovative aspects of this research, as compared with our previous paper.

After our study was published (2014), the introduction of new high-resolution mass spectrometry (i.e. the Orbitrap Velos) and of new quantitative bioinformatics tools permitted us to realize the first quantitative high-resolution proteomic study in seminal plasma samples of patients affected by s HH, before and after only 3 months of TRT.

The evaluation has been performed after 3 months of TRT because it is the proper timing of response to exogenous androgen administration in terms of restoration of seminal vesicle size, which is a sensitive measure of androgen genomic effect.

The improvement in technological and bioinformatics tools permitted now to observe not only the identification of the protein after 6 months, as in the previous study, but the early modulation of the protein quantity, which was observed just at 3 months of TRT.

We discussed these aspects in the manuscript and modified the manuscript according to your evaluable suggestions.

A native English speaker revised the text for written English and approved the manuscript.

Introduction

L80: please be more precise and explain the molecular mechanism of what you thought about?

L95: the aim of study should be more precise

The text has been amended to take in along with your valuable suggestions.

Material and methods

Materials and methods should be written in more details.

L120: Was the effectiveness of TRT measured?

Testosterone and estradiol assay have been performed to verify the effectiveness of TRT.

L182-191 Was the abundance of identified proteins significantly different?

L186, 190, 191: why different ratio was applied for comparison HH and healthy males (ratio>1.5 less abundant for HH and <0.67 more abundant for HH) and before and after TRT (ratio>1.5 under-expressed after TRT and <0.67 over-expressed after TRT). Usually ratio below above 2 is considered for proteomics.

Since we had only 10 samples per group we did not perform statistical analysis for protein abundance, but analyzed the ratios between groups as previously performed in several proteomic papers in literature . Furthermore the cut-off values for the ratio for considering a protein as over-expressed or under-expressed have not been standardized. Some papers reported cut-off values of about 1.5 and 0.67 (Xu, 2019; Grande 2018) although several ones have been published with less stringent cut-off values. For example, Gong SN in 2018 (World J Gastroenterol) used the cut-off values of 1.2 and 0.83; Zhang in 2018 (Biomed Res Int) reported cut-off values of 1.3 and 0.67;Jiand used in 2017 (Sci Rep) values of 1.2 and 0.8.

We applied so cut-off values previously used in literature, which, if we review the proteomic literature, might be considered as quite stringent quantitative cut-off values.

L201: there is no information about additional 7 clinical and hormonal hypogonadic patients samples used for western blot.

We reported in Supplementary Table 1 the clinical, hormonal and seminal data of the additional 7 patients which have been added for Western blot analysis

L205-213: amount of protein loaded on SDS-PAGE is unknown.

L212-213: please add reference for red ponceau staining as internal standard

L196-213: there is lack of description of statistical method that were used for analysis of data presented in Fig.1

We modified the manuscript according to your suggestions. We improved the manuscript in explaining these points.

Results

The results seems to be incomplete and at least some hypotheses should be tested.

Table 1. there are only results concerning hypogonadic patients and control. Why authors have not provided the same data for patients before and after TRT.

We added a table for testosterone and estradiol level after TRT, to confirm the effectiveness of TRT.

L227: what does it mean "the adoption of stringent protein identification criteria". Please, describe all criteria in details in M&M section.

The text has been amended to take in along with your valuable suggestions.

It means that from more than 100 identified proteins (L224 and 225) only 11 was differentially expressed between HH and healthy patient. And all of them was lower for HH, there was any protein characteristic for HH?

The sentence at L224-225 was added to specify why we identify with stringent criteria a small amount of proteins per sample. In fact, we identified – before filtering – 110-176 proteins per sample, and semenogelins resulted in the majority of the identified spectra.

As a consequence, after applying stringent filtering criteria, we observed only 12-60 proteins per sample (12-17 in patients and 15-60 in controls) which have been identified with high accuracy. Among these proteins, 11 proteins have been observed as reduced in HH patients, which were the ones considered for this study. No proteins were observed exclusively expressed in the group of patients or increased in patients.

The same for table 3. Any proteins with higher abundance before treatment.

We emended the manuscript and specified that no proteins have been observed at higher abundance before treatment.

Fig. 1. In the supplementary material I would like to see the picture of all gels after enhanced chemiluminescence, all pictures of membranes after red ponceau staining and western blot membrane with detected bands. Please show the calculation and complete the description of the charts (Y axis is empty).

In the supplementary material section, we reported the picture of all gels after enhanced chemiluminescence, all pictures of membranes after red ponceau staining and western blot membrane with detected bands. Quantification has been furthermore reported as supplementary material. We moreover modified the figure according to your evaluable suggestions.

Discussion

L:263-283. Do you suggest homeostasis disorder between proteases and their inhibitor in HH patients?

We clearly reported the suggestion deriving by our data of an imbalance between preoteases and protease inhibitor in seminal plasma of HH patients.

L274: You mentioned about malformed sperm for mice, could you check your hypothesis (L282-283) for HH patients?

We modified the manuscript as following: “The reduction of PCI in hypogonadic patients might reflect an impairment in spermatogenesis or might represent a signature of defective spermatogenesis in HH patients, as well as in epididymal function. Further studies are needed to confirm these hypotheses.”

L292-293, could you support your hypothesis concerning reduction of defence mechanism against infection transmission during sexual intercourse in HH patients.

We discussed this topic according to your evaluable suggestion. Previous data have been reported in literature demonstrating that HH patients has higher incidence of prostatitis and male tract infection/inflammation (MAGI) (Condorelli, 2014) and increased levels of MAGI markers in seminal plasma (Milardi, 2018). The observation of a reduction in gastricsin in our study might represent so a molecular by which hypogonadic patients have more frequently male tract infections. Further studies are needed to confirm this hypothesis.

L318. To be sure that the level of some proteins were restored you should compare HH patients after TRT with control. Now you can only suggest the increase of proteins after TRT.

The text has been amended to take in along with your valuable suggestion.

Round 2

Reviewer 1 Report

Accept after minor revision

Reviewer 2 Report

The authors have revised manuscript according to my comments.

This manuscript is a resubmission of an earlier submission. The following is a list of the peer review reports and author responses from that submission.